## PERSPECTIVE

# Reassessing a cryptic history of early trilobite evolution

James D. Holmes [1✉] & Graham E. Budd [1]

Trilobites are an iconic Paleozoic group of biomineralizing marine euarthropods that appear abruptly in the fossil record (c. 521 million years ago) during the Cambrian 'explosion'. This sudden appearance has proven controversial ever since Darwin puzzled over the lack of pre-trilobitic fossils in the *Origin of Species*, and it has generally been assumed that trilobites must have an unobserved cryptic evolutionary history reaching back into the Precambrian. Here we review the assumptions behind this model, and suggest that a cryptic history creates significant difficulties, including the invocation of rampant convergent evolution of biomineralized structures and the abandonment of the synapomorphies uniting the clade. We show that a vicariance explanation for early Cambrian trilobite palaeobiogeographic patterns is inconsistent with factors controlling extant marine invertebrate distributions, including the increasingly-recognized importance of long-distance dispersal. We suggest that survivorship bias may explain the initial rapid diversification of trilobites, and conclude that the group's appearance at c. 521 Ma closely reflects their evolutionary origins.

The timing of early animal evolution is a controversial problem, with opinion generally divided between two major schools of thought. One view holds that the fossil record is relatively accurate, and that the appearance of the major animal groups during the Cambrian radiation closely reflects their evolutionary origins[1–3]. The opposing view, based largely on divergence dates estimated by molecular clocks, suggests a much deeper evolutionary history[4–7]. In recent years, differences between these views have reduced[8], with recognition of certain members of the Ediacara biota as probable eumetazoans[9,10] and more refined molecular divergence estimates coming closer in-line with the fossil record[6,11]. Nevertheless, there remains a mismatch between these views, particularly as to whether the classic 'explosion' of taxa (including trace fossils) across the Cambrian Terreneuvian Series represents the true radiation of bilaterian crown groups, or whether the origins of these are to be found earlier, in the Ediacaran.

Trilobites are a clade of total-group euarthropods whose first appearance datum (FAD) marks the boundary between the Terreneuvian and provisional Cambrian Series 2 (currently dated to c. 521 Ma)[12,13]. They are one of the largest and most successful Paleozoic groups, persisting for some 270 million years, and represented by over 22,000 described species[14,15]. This excellent fossil record—a result of their easily-preserved, biomineralized exoskeleton that was molted many times during life—can be used to address important questions concerning early animal evolution[16,17]. Trilobites have been viewed as exemplary for the argument of deep bilaterian (and therefore metazoan) divergence dates, and formed an important part of the argument for early proponents of this view[4,18,19]. One reason for this is that trilobites supposedly show substantial provincialism when they appear in the fossil record, being separated into two major biogeographic areas in the early Cambrian: the 'olenelline' province (e.g., Laurentia, Baltica) and the 'redlichiine' province of Gondwana (including Antarctica, Australia, China and India, amongst other regions), with a transitional zone (sometimes referred to as the 'bigotinid' province) occurring in areas such as West Gondwana and Siberia[20–22] (Fig. 1; we use suborders here[23]). It

[1] Department of Earth Sciences, Palaeobiology, Uppsala University, Villavägen 16, Uppsala 752 36, Sweden. ✉email: james.holmes@geo.uu.se

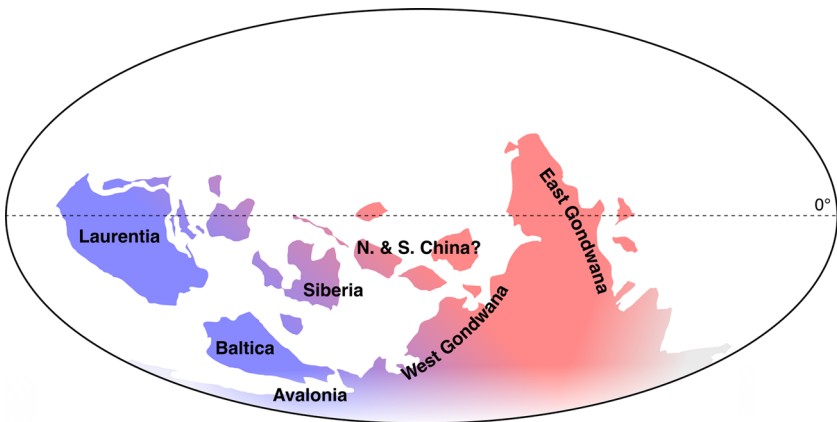

**Fig. 1 Global palaeogeographical reconstruction at c. 520 Ma showing hypothetical locations of palaeocontinents and trilobite faunal provinces.** Colors show the approximate areas of the so-called 'olenelline' (blue) and 'redlichiine' (red) early Cambrian trilobite faunal provinces, and the transitional (purple) area between these. Importantly, this provincialism is not strongly evident in the earliest trilobite faunas, which are known from the transitional areas of Siberia and West Gondwana, as well as Laurentia (see main text for details). Note that the positions of North (N.) and South (S.) China are consistently uncertain in Cambrian geographic reconstructions and are marked as such (?). Map based on fig. 2.7 of Torsvik & Cocks[92].

has generally been assumed that trilobites must have a cryptic evolutionary history for such a pattern to be produced, and that observed distributions of taxa are a result of vicariance, in this case resulting from supercontinent breakup and the subsequent isolation of certain paleocontinents. These patterns have been linked to either the breakup of Rodinia (c. 700–800 Ma)[19] or the ephemeral Pannotia (c. 550–600 Ma)[24–26], although the refinement of molecular clock estimates suggests that the former in particular is unlikely. Given the accepted position of trilobites as total-group euarthropods[27], linking of these biogeographic patterns to supercontinent breakup in the Neoproterozoic has been used to support the argument for a deep, cryptic history of arthropod evolution, and early animal evolution more generally[4,18,19,24,25]. However, this reasoning is based on two major assumptions: (a) that the earliest trilobites already show established biogeographic provincialism and phylogenetic diversity; and (b) that observed biogeographic patterns result from vicariance rather than dispersal. It also raises questions about the early trilobite fossil record that are not easily answered. For example, if a biomineralized exoskeleton and associated traits are synapomorphies of the group, why are Terreneuvian trilobites absent from the fossil record despite an adequate shelly record across the same period?

Here we review the evidence for a cryptic history of early trilobite evolution. We also review the assumptions behind the supporting vicariance model, in light of >30 years of work that has greatly improved our understanding of the factors responsible for modern biogeographic patterns, including the relative importance of vicariance and dispersal.

**The biomineralization problem.** Although alternative views have been proposed[28,29], trilobites (probably excluding agnostines[30]) are now almost universally accepted to form a clade defined by several synapomorphies, including calcification of the dorsal exoskeleton, and eyes with calcified lenses and circumocular sutures[31–33]. Dorsal facial sutures may also be a synapomorphy of the group, either excluding or including olenellines depending on whether their absence in this group represents the ancestral state or secondary loss respectively (dorsal facial sutures may also be independently derived in eodiscids[16]). As such, any ancestral trilobite that lived prior to their FAD (at c. 521 Ma) should possess a biomineralized exoskeleton (unless secondarily lost); however, no Terreneuvian trilobite has been found despite the presence of diverse shelly

faunas across much of this period. Any argument that Terreneuvian (or Precambrian) trilobites are absent from the record because they were non-mineralized implies that all of the major trilobite lineages must have diverged sometime during this earlier period, and then all independently evolved a calcitic exoskeleton at about c. 521 Ma or shortly thereafter. This argument effectively discards the defining synapomorphies of the clade, implying that trilobites are polyphyletic as currently conceived— a view that most workers would reject. Arguments have been made for such rampant convergence, e.g., changes in seawater chemistry triggering the development of calcitic skeletons in several bilaterian lineages around this time[16]. However, it remains highly improbable that the characters associated with biomineralization mentioned above were developed independently in all lineages hypothesized to cross this boundary. For example, based on the recent Cambrian trilobite phylogenetic analysis of Paterson et al.[16] this implies eleven independent acquisitions of a calcified exoskeleton and associated eye structures, and seven acquisitions of dorsal facial sutures (based on their multi-epoch clock model and ignoring divergences from 521 to 522 Ma to allow for uncertainty in the record; see their fig. 1[16]). In addition, if trilobites as currently conceived really did independently evolve these traits, this implies the presence of a large number of non-mineralized trilobite sister groups at this time, and one would expect to see some of these in the various early Cambrian *Lagerstätten*. Yet, there are no examples from these deposits that can be considered a 'non-mineralized trilobite' (note that any such taxon would have to be more closely related to a particular trilobite group than to other trilobites). It seems highly unlikely that all of these suddenly went extinct in the interval between c. 521 Ma and the oldest of these deposits (e.g., the Chengjiang biota), or for some reason are not preserved.

In the absence of any other compelling reason to doubt the accuracy of the trilobite fossil record, it is thus much more parsimonious to assume that the characters discussed above represent true synapomorphies, and that trilobites arose close to their FAD. It should be noted that although the position of agnostines does not greatly affect the arguments presented here, the exclusion of this group from Trilobita suggests a possible independent acquisition of a calcified dorsal exoskeleton.

**Arthropod traces in the Terreneuvian.** Trace fossils have also been used to argue for a cryptic history of trilobite evolution. Arthropod traces from the Terreneuvian such as *Rusophycus* have

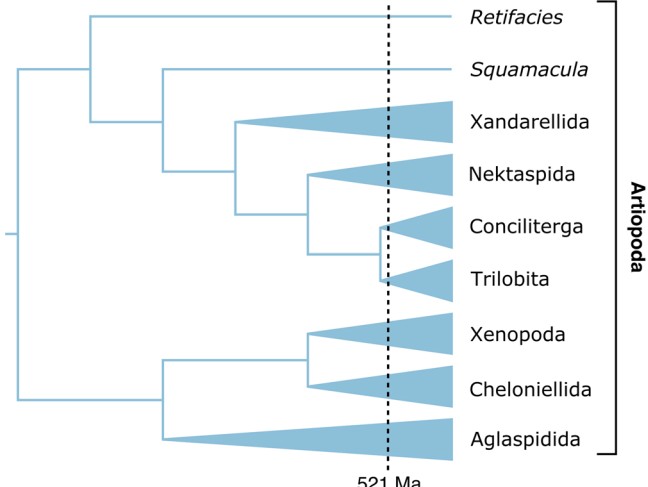

**Fig. 2 Simplified phylogeny of the Artiopoda.** A uniting feature of this group are their similar biramous appendages and it is likely that some of the non-trilobite artiopodans produced the same kinds of traces as trilobites (e.g., *Rusophyscus*, *Cruziana*). If trilobites arose close to when they appear in the fossil record (c. 521 Ma) there must have been a substantial earlier history of artiopodans (and other more distantly related taxa), thus obviating the requirement to suggest trilobites produced these traces in the Terreneuvian. Topology based on fig. 6b of Ortega-Hernández et al.[38].

been attributed to trilobites[34] (despite appearing c. 10–15 million years prior to the first appearance of the group[35]), although some authors have since suggested that these could have been produced by other arthropods[36,37]. Trilobites are members of a much larger diversity of Cambrian euarthropods, the great majority of which are non-mineralized. Many of these (like trilobites) exhibit a series of biramous ('two-branched'), gnathobase-bearing appendages along the anterior-posterior axis[38–40]. In particular, the Artiopoda (a large clade of trilobite-like euarthropods, including trilobites) generally have very similar appendages[38], and this is clearly a primitive trait of the group. Many artiopodans also exhibit comparable overall morphologies to trilobites, and some of these likely filled similar ecological niches. Thus, it might be expected that such taxa produced similar traces across the early history of artiopodans, which must have occurred prior to the FAD of trilobites (Fig. 2). It is even possible that more basal non-mineralized stem-euarthropods (e.g., fuxianhuiids[41] and *Parapeytoia*[42], which exhibit the same basic appendage structure) could produce similar traces. The recent interpretation of Cambrian Series 2 (Stage 4) *Rusophycus* from Canada as being produced by a non-mineralized crustacean-like arthropod[43] supports the idea that these early traces could be produced by non-trilobites. It has also been pointed out that although trace fossils like *Rusophycus* and *Cruziana* occur after the Permian mass extinction (e.g., in the Triassic[44,45]), this is not considered evidence of post-Permian trilobites[46] (these are also attributed to crustacean-like taxa). Why should we consider the presence of these traces prior to c. 521 Ma in a different light, when other obvious candidates for producing them are present? Based on the above, a more literal reading of the trilobite fossil record is not incongruent with the trace fossil record. Rather, it supports the interpretation of these traces representing the early diversification of total-group euarthropods starting in the early Terreneuvian (Fortunian), more derived artiopodan-type taxa later in the Terreneuvian (e.g., the more 'typical' *Rusophycus* occurring in Stage 2[35]) and allows additional time for the evolution of trilobites before their FAD at c. 521 Ma.

**The vicariance assumption.** Distributions of related terrestrial and freshwater taxa separated by ocean basins can be explained by either vicariance resulting from continental separation, or oceanic dispersal (i.e., long-distance dispersal across oceans). In the *Origin of Species*[47], Darwin argued that occasional dispersal events—rare but occurring regularly across vast periods of geological time—could explain such distributions, and undertook various experiments to show that certain plant seeds can remain viable after long periods immersed in salt water, or in the crops and digestive tracts of birds. He essentially argued against vicariance-only explanations by suggesting that dispersal of successful organisms likely explained the observation that fossil faunas were most similar within the same intervals of geological time (even across great distances), and that this was supported by the marine faunas of different continents being more similar than terrestrial faunas[47]. Darwin also noted that organisms on volcanic islands must have arrived by dispersal, and this remains a strong argument today—the important of such events in populating oceanic islands on geological timescales cannot be denied[48,49]. However, the validation of plate tectonics and the rise of cladistics from about the 1960s onwards provided a compelling model of vicariance by continental fragmentation testable with phylogenetic hypotheses, and vicariance (or cladistic) biogeography became the dominant paradigm[50]. In contrast, hypotheses of oceanic (or what Darwin referred to as 'accidental') dispersal were considered unscientific as they could not be falsified, and essentially every pattern could be explained by invoking some series of dispersal events[50] (although as Darwin pointed out, factors controlling dispersal such as ocean currents and prevailing winds are not 'accidental'[47]). Nevertheless, it was under the general assumption that vicariance must be responsible that such explanations for trilobite biogeographic patterns emerged.

Over the last 25 years or so, it has become clear that oceanic dispersal plays a much greater role in explaining modern biogeographic distributions than previously thought[50–52], and methods of vicariance biogeography that a priori exclude dispersal as a factor have been widely criticized[53–55]. Molecular dating has shown that phylogenetic splits within many groups thought to represent instances of vicariance by continental separation occurred much too recently for this to be the case[50]. Disjunct distributions will often be controlled to some extent by tectonic movements, but this recent work suggests that vicariant signal is often overlaid to a greater-or-lesser extent by instances of oceanic dispersal, creating a complex web of patterns that varies in different groups[56]. This is true even for groups once considered to show classic Gondwanan vicariant distributions (i.e., related to the breakup of Africa, Antarctica, Australia, India and South America), including cichlids and other freshwater fish[57,58], and various plant groups including *Araucaria* and *Nothofagus*[56,59]. This does not mean that vicariance is an insignificant factor in explaining biogeographic distributions, and it is still suggested to be the dominant factor in some cases (e.g., in Southern Hemisphere terrestrial animals[56]). However, it is now clear that we can no longer simply assume that vicariance is the major factor controlling such distributions.

Early work proposing vicariance as the major factor in explaining early Cambrian trilobite biogeographical patterns[4,18,19] stated that provincialism shown by the earliest trilobites essentially proved an "earlier phase of vicariance unrecorded in the fossil record" (Fortey et al.[19], p. 18). Coupled with the implicit assumption in this reasoning that vicariance must be the cause of such patterns (rather than dispersal), the extremely deep divergence dates being proposed at this time by early molecular clocks suggested that trilobites may have a cryptic history extending back several hundred million years, and that

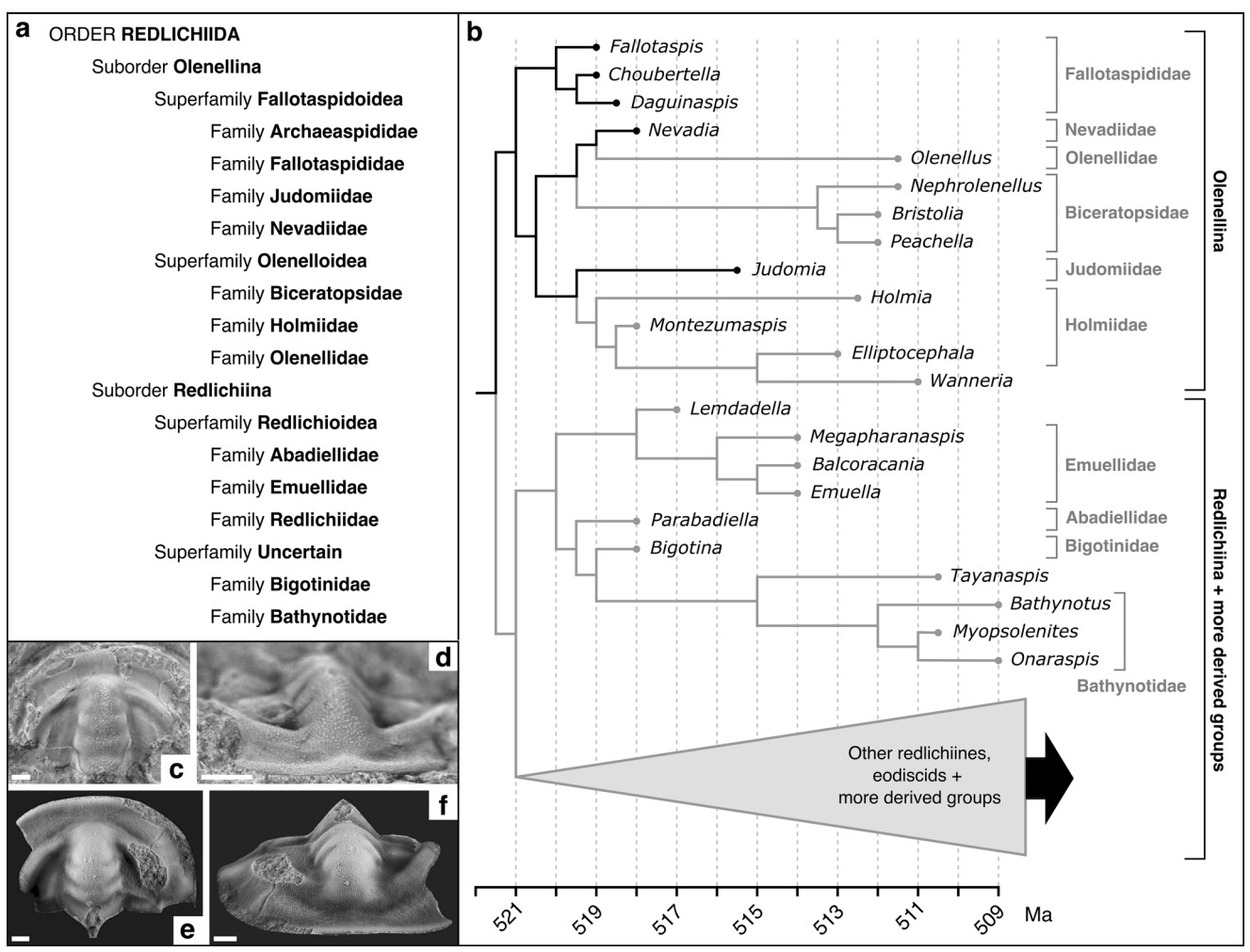

**Fig. 3 Relationships between early trilobites in the Order Redlichiida. a** Higher-level taxonomy of the trilobite Order Redlichida, largely after Adrain[23]. Note that this does not represent an exhaustive list of taxa within the various groups shown, only those discussed in the text or shown in **b**. **b** Phylogeny of early Cambrian trilobites (modified from fig. 2 of Paterson et al.[16]; see this publication for full species names and dating references). Tips represent ages of included species, but although we have attempted to retain approximate relative internal branch lengths (while increasing rates towards the base of the tree), these are not precisely to scale. The tree is rooted between the suborders Olenellina and Redlichiina (see main text) with a divergence date of 521.5 Ma, to illustrate the scenario of a relatively accurate fossil record. Comparison between the early redlichiine bigotinid *Bigotina bivallata* (**c, d**) and the early olenelline fallotaspidoid *Profallotaspis tyusserica* (**e, f**). These taxa show considerable morphological similarities, supporting a close relationship between these groups. Modified with permission from Bushuev et al.[78] and Geyer;[80] scale bars 1 mm.

vicariance associated with the breakup of Rodinia might be responsible for the observed biogeographic patterns.

More recently, several studies that used a modified version of Brooks Parsimony Analysis have suggested that trilobite biogeographic patterns may have resulted from the breakup of the more recent, ephemeral (and controversial[60,61]) supercontinent Pannotia in the late Neoproterozoic[24,26], in line with more recent molecular clock estimates[6,11]. These methods involve replacing terminal taxa and internal nodes in phylogenies with biogeographic distributions, and coding the resulting information from these in a separate character matrix, where the 'taxa' represent biogeographic regions[62]. This matrix is then subjected to a parsimony analysis and the resulting tree topologies are interpreted to reflect histories of faunal connections (in this case between palaeocontinents). 'Vicariance' trees are meant to reflect branching associated with the erection of barriers between areas, whereas 'geo-dispersal' trees reflect the removal of barriers[24,62]. As such, it is important to note that geo-dispersal represents a completely different concept to oceanic dispersal. In fact, there is no mechanism within these models that allows for true oceanic

dispersal, in that any relationships resolved are interpreted to result from isolation or amalgamation of areas due to the erection or removal of barriers between these (e.g., related to tectonic movements or sea-level fluctuations). Therefore, such models can produce well-resolved 'vicariance' trees that are not necessarily the result of vicariance. For example, similar histories of connections in different groups can be produced by dispersal influenced by factors such as relative distances between continents and ocean currents[50], which are primary controls on modern marine animal distributions. As mentioned above, there are many modern cases where molecular dating supports explanations of oceanic dispersal, despite phylogenetic relationships apparently being consistent with vicariant hypotheses[50]. Clearly then, timing of divergences is a key issue when interpreting biogeographic patterns in light of dispersal and vicariance. Recent morphological clock estimates that suggest trilobites most likely arose in the Fortunian[16] (likely over-estimates; see below) represent a situation analogous to these modern cases, with the age of the group apparently post-dating estimates of continental separation. As with these modern

examples, if a vicariance explanation is shown to be inconsistent due to a mismatch in timing between tectonic events and taxon divergence estimates, then oceanic dispersal is likely to be responsible.

**Dispersal: a major factor in explaining marine invertebrate distributions.** The argument over the importance of vicariance (by continental separation) versus oceanic dispersal in explaining modern biogeographic distributions may be largely irrelevant to the case of trilobites, as this form of vicariance has generally only been used to explain distributions of terrestrial and freshwater taxa[50]. In modern marine animals—including taxa restricted to continental shelves as most early trilobites were—instances of vicariance almost always involve the splitting of populations by land (rather than by seaways as invoked for Cambrian trilobites)[63–65]. Today, marine invertebrate distributions—including differences in faunas separated by deep ocean basins (e.g., Atlantic, Pacific)—are controlled by varying dispersal ability in relation to a complex series of factors, including ocean currents, physical barriers and latitudinal/temperature gradients, rather than relictual vicariant patterns from when these seaways first opened[66,67]. Thus, invoking continental separation as the major factor in explaining trilobite distributions is inconsistent with what is known for the modern marine fauna.

Rapid dispersal (geologically speaking) has been considered less likely in explaining Cambrian trilobite biogeographic patterns than a cryptic history requiring invocations of rampant convergence or rarity[16]. In fact, marine invertebrates can generally disperse rapidly across large distances[66], either by planktonic larval dispersal, or other measures almost certainly present during the Cambrian, such as transport on semi-submerged substrates such as floating macroalgal or pumice rafts[68,69]. For example, even in the last few hundred years, pumice rafts have occurred frequently in all major oceans, and have been shown to develop diverse assemblages and transfer organisms vast distances (e.g., in a recent incident >80 species and >5000 km[70]). Southern Hemisphere coastal communities are also biologically linked by frequent, long-distance dispersal on macroalgal rafts[69]. Such rafts have helped to confirm that marine invertebrate distributions are strongly controlled by ocean currents[70–72]. Importantly, although the majority of recruitment to such vectors is by taxa with pelagic larvae, benthic marine invertebrates lacking such stages are also transported regularly on both pumice and macroalgal rafts[68,73,74]. It is reasonable to assume that trilobites (particularly small individuals) could be as well, and that dispersal was a major driver of marine invertebrate distributions in the Cambrian, as it is today.

**(A lack of) provincialism and phylogenetic diversity in the earliest trilobites.** It has been stated repeatedly that, even in their earliest history, trilobites show marked biogeographic provincialism and phylogenetic diversity[4,16,18–20,75], being separated into the 'olenelline' and 'redlichiine' provinces discussed previously. If indeed trilobites show distinct provincialism from the point they appear in the fossil record, then this might be considered a strong argument for a cryptic history. However, even a brief examination of the data shows that this view is not well supported.

It seems clear that the earliest trilobites appear at a similar time in Laurentia, Siberia and West Gondwana[76,77], and initially consist of olenelline fallotaspidoids from the families Archaeaspididae and Fallotaspididae, and redlichiines from the Family Bigotinidae[78–82] (Fig. 3a), which have been suggested to be closely related[80]. Despite the argument for provincialism, similar fallotaspidoids are present in all three of these regions, while bigotinids are apparently absent from Laurentia. The oldest

trilobites in China and East Gondwana are primitive redlichioids such as *Parabadiella*, the ages of which are somewhat controversial[76,77,83]. In Australia, it has recently been suggested that these appear at about the same time as the oldest trilobites from the regions discussed above;[84,85] however, these East Gondwanan trilobites show similarities to redlichioids found slightly higher in the West Gondwanan successions[86]. Regardless, these clearly represent primitive forms likely closely related to bigotinids[16,80]. In Baltica and Avalonia, the first trilobites to appear are derived olenelline holmiids generally thought to be somewhat younger than the taxa discussed above[76].

Based on this, it is clear that faunas occurring within the lowermost several trilobite biozones (representing perhaps several million years) in regions where trilobites first appear are similar, and contain primitive forms that are more-or-less closely related —in direct contradiction to the claim of 'established provincialism' in the earliest trilobites. Only after this time did the relatively distinct faunas of the two major biogeographic provinces become established, essentially involving a radiation of redlichiines in Gondwana (and elsewhere), and a continued dominance of olenellines in Laurentia. However, there is clear overlap of these high-level taxa, with sutured trilobites appearing rapidly in all regions. It has been suggested previously that such patterns may result from the continued rifting seen in the early Cambrian (following Neoproterozoic breakup), with dispersal between palaeocontinents initially being easier, and faunas becoming more provincial later as continental separation continued[87]. This illustrates how distributions can reflect tectonic relationships without requiring strict vicariance by continental fragmentation. Worldwide trilobite biogeographic patterns in the early Cambrian have not actually been explored in great detail (although see Álvaro et al.[75]) and should be more thoroughly investigated. For example, overlap of elements within the 'distinct' faunas discussed above suggests that such patterns might be better explained by a more continuous longitudinal gradient, ranging from Laurentia in the west to East Gondwana in the east (Fig. 1).

The claim that the earliest trilobites show established phylogenetic diversity also seems to have little basis. As mentioned above, the earliest trilobites to appear are olenelline fallotaspidoids and redlichiine bigotinids, and despite these groups being classified in different suborders based largely on the presence (Redlichiina) or absence (Olenellina) of dorsal facial sutures, these groups show marked morphological similarities. For example, a comparison of the bigotinid *Bigotina bivallata* (Fig. 3c, d) and the fallotaspidoid *Profallotaspis tyusserica* (Fig. 3e, f) shows comparable overall proportions, and considerable similarities in the shape and orientation of the eye ridges, and how these extend from the anterior of the glabella—this seems to be a primitive trait of the earliest trilobites[80]. It has generally been assumed that early olenellines lacking dorsal facial sutures (such as fallotaspidoids) represent the ancestral state. We agree that this is the most parsimonious (although not the only) alternative, and the presence of anterior facial sutures in the olenelline *P. tyusserica* (Fig. 3e, f) may provide a clue as to how dorsal sutures first evolved in trilobites[78]. Whilst posterior facial sutures appear to be absent in this species, opening of the anterior sutures (linking the circumocular and perrostral/marginal sutures) would have substantially widened the ecdysial gape, and also made it much easier for the 'free' cheeks to break off during molting, as appears to have happened in the specimen refigured here (Fig. 3e, f). The presence of anterior facial sutures in *P. tyusserica*, as well as the very similar overall form of this species and bigotinids such as *B. bivallata*, suggests that these morphologies may be close to the split between the sutured redlichiines and unsutured olenellines, with the earliest representatives of both groups essentially showing continuous variation through these forms.

Thus, the separation of the earliest redlichiines and olenellines into different suborders based largely on the presence/absence of dorsal facial sutures substantially overstates the amount of morphological variation present in the earliest trilobites. This illustrates the unavoidable limitations of the taxonomic system when splitting high-level taxa close to their origins (we must 'draw the line' somewhere).

**Diversification through time and morphospace**. An important consideration of a more literal interpretation of the trilobite fossil record is the consistency of this model with hypothesized phylogenetic relationships and observed patterns of diversity through time. Figure 3b shows a simplified version of a recent Cambrian trilobite phylogeny (based on the 'relaxed clock' model of Paterson et al.[16]), with a (modified) divergence date of 521.5 Ma —illustrating a scenario of a relatively accurate fossil record. Under this assumption, and given the likely primitive absence of dorsal facial sutures, the rooting of the original tree on the Fallotaspididae is reasonable. However, given that the oldest fallotaspidoids (not included in the analysis, e.g., *Profallotaspis*) tend to show morphologies more intermediate with bigotinids, an argument could be made to root the tree closer to the split between these groups, and this is what is shown in Fig. 3b. This makes little difference to the arguments presented here, although it does illustrate a very early split between the Olenellina and Redlichiina, as might be expected based on the fossil record. In either case, the Fallotaspidoidea are suggested to be paraphyletic (darker branches at the base of the tree), giving rise to a polyphyletic Olenelloidea. This supports the idea of a fallotaspidoid-type morphology being a primitive trait, present across the base of the tree. It also supports the early divergence of primitive redlichioids (e.g., bigotinids, abadiellids) from a fallotaspidoid-type morphology (we note that under the 'epoch clock' model[16] these are suggested to be slightly more derived).

If trilobites have a substantial cryptic history, we might expect to see different morphologies appearing randomly in time and morphospace, as previously invisible lineages appear in the fossil record. However, in general, it seems that stratigraphic appearance closely reflects phylogeny, suggesting that this is not the case. Initially, we see a very small number of related families, followed by the appearance of more derived groups interpreted as diversifying from these older taxa. For example, there is a clear gradient of morphologies from fallotaspidoids, through bigotinids[80] and early redlichioids such as *Lemdadella*[88] and *Parabadiella*[83], to younger, more derived forms such as *Eoredlichia*[89] and *Redlichia*[39]. The same is true for olenellines: from early fallotaspidoids, through later examples like *Nevadia* and *Judomia*, to younger olenelloids like *Olenellus* and *Holmia*[82] (Fig. 3b). Other early groups like the ellipsocephaloids are likely to have evolved from early redlichioid or bigotinid-type morphologies (the Bigotinidae are sometimes included in this group[23]). Such patterns of appearance and diversification through time suggest that we are observing a real radiation, and that there is no need to invoke an extended cryptic evolutionary history to explain trilobite biogeographic distributions or phylogenetic relationships in the early Cambrian.

**The push of the past**. Recent studies[90,91] have emphasized that successful clades that survive for long periods of time are statistically likely to have high initial rates of diversification, an effect termed the push of the past (POTPa). Such clades have a higher chance of surviving for any given length of time compared to those with low initial diversification rates. The size of the POTPa is controlled by turnover rates; i.e., for a given rate of diversification (speciation-extinction rate), a higher POTPa is implied for

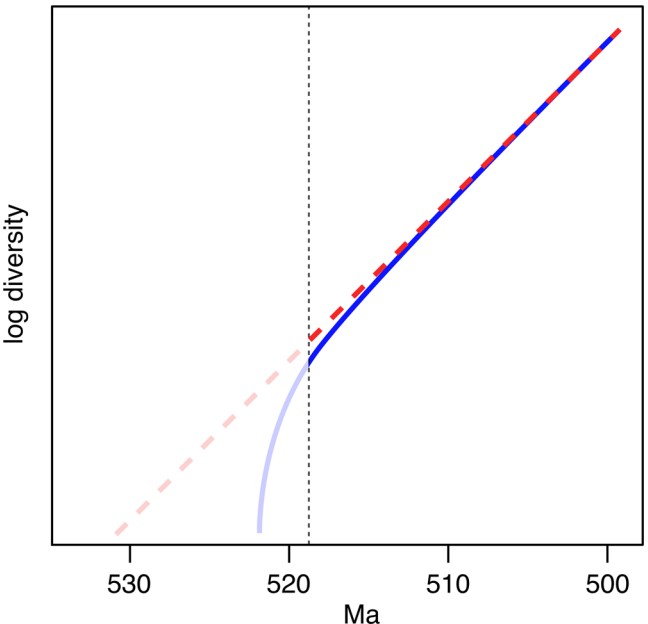

**Fig. 4 An illustration of the likely effect of the push of the past (POTPa) on the early radiation of trilobites.** The solid blue line shows a scenario with a substantial POTPa, as might be expected for an extremely diverse and successful clade like Trilobita. The diversification rate (the slope of the line) is initially very high before falling to the background rate. The dashed red line shows a projection of trilobite diversity into the unobserved early history of the group based on the 'visible' part of the tree—in this case after 519 Ma (shown by the vertical dashed line), similar to the Cambrian trilobite phylogenies and morphological clock divergence estimates of Paterson et al.[16]. Survivorship biases like the POTPa may explain the discrepancies between such estimates and the trilobite fossil record.

a high extinction rate. Such an effect is likely to significantly shorten the cryptic history predicted by recent morphological clocks[16], which suggested that trilobites most likely arose in the Fortunian. These authors noted that their projections were necessarily based on rates within the visible part of the tree (after 519 Ma, the age of the oldest trilobites included in their analysis), and that if rates were higher prior to this then the estimated divergence dates would be overestimates[16]. They also showed that enforcing a divergence date of 522 Ma would require elevated rates at the base of the trees (as expected by the POTPa), but this had little effect on their overall results; rates remained relatively homogeneous across the Cambrian[16]. This suggests any reversion to the background rate occurred quickly, and that the slightly higher rates they observed in the early Cambrian may result from a short period of rapid diversification close to when trilobites appear. As such, we suggest that the mismatch between their divergence estimates (within error bars as young as c. 526 Ma) and an even more literal reading of the trilobite fossil record are potentially explained by effects such as the POTPa (Fig. 4). It should also be noted that the POTPa suggests that it is unlikely for the absence of trilobites in the Terreneuvian to be explained by rarity or low diversity, as such groups are highly unlikely to persist for long without going extinct[90,91]. In fact, the effect of the POTPa predicts that diversity will initially increase rapidly, meaning that a fossil group like trilobites with high preservation potential should appear in the record soon after it arises.

## Conclusions
The suggestion of non-mineralized trilobites in the Terreneuvian or earlier is shown to be highly unparsimonious, implying

rampant convergence of structures associated with exoskeletal biomineralization in all major early trilobite lineages, and abandonment of the synapomorphies uniting the clade. This suggests that no credible reason has been proposed for the absence of Terreneuvian trilobites in the fossil record, given the assumption of a substantial cryptic evolutionary history. Despite previous statements to the contrary, when trilobites appear in the fossil record they show limited provincialism and relatively low phylogenetic diversity. Even when more distinct faunas develop across the remainder of Cambrian Series 2 there is considerable overlap between these, and patterns of diversification suggest this is occurring in real time (rather than resulting from divergence prior to the FAD of trilobites). Given the change in our understanding of the relative importance of vicariance and dispersal in explaining modern biogeographic patterns over the last several decades—and the general observation that modern marine invertebrate faunas do not show vicariant patterns resulting from continental separation—trilobite biogeographic patterns are unlikely to result from this form of vicariance. The mismatch between recent morphological clock estimates (that suggest trilobites probably emerged in the Fortunian) and an even more literal reading of the fossil record can be explained by effects such as the push of the past, which anticipates higher rates of diversification during the initial radiation of clades—particularly in the case of very long-lived and successful groups like trilobites. We conclude that the FAD of trilobites closely reflects their evolutionary origins, and that there is no compelling evidence to suggest an extended cryptic evolutionary history for this group.

**Reporting summary**. Further information on experimental design is available in the Nature Research Reporting Summary linked to this paper.

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

## Acknowledgements
We thank Nigel Hughes and Greg Edgecombe for useful comments that substantially improved the manuscript. We also thank Marissa Betts for comments on an early version of this work, and Ben Slater for insightful discussions.

## Author contributions
J.D.H. conceived the study and wrote the manuscript with input from G.E.B.

## Funding

## Competing interests
The authors declare no competing interests.
