## [Peer Review File · Communications Biology]

Reviewers' comments:

Reviewer #1 (Remarks to the Author):

Review of: Reassessing a cryptic history of early trilobite evolution

Authors: Holmes and Budd

Reviewer: Nigel Hughes

Overall recommendation: Accept with relatively modest modification

This is a well written piece of general interest to evolutionary biologists because it touches on an important general issue – vicariance vs dispersal, and is specifically aimed at an important issue in paleontology – the tempo and mode of the Cambrian radiation. I have long been sympathetic to the general argument made herein and I think the authors have done a good job of marshaling their various arguments in a cogent and interesting way that's an easy read for the non-specialist. I might imagine this piece fits well under the title of "perspective".

Major comments

That fallotaspidoids and bigotinids are close relatives is nicely illustrated in Fig 2, as is the discussion about the anterior facial suture in the *Profallotaspis*. I can accept this argument. However, the editor will have to judge whether this illustration alone is enough to make the point sufficiently for a "perspective" piece. I don't think it would be enough on its own for a research report, in which we'd want to see the arguments made in the text supported by a phylogenetic analysis. I think the paper would certainly be stronger if a character-based analysis could be presented or referred to. Without it the paper seems to go 90% of the way, but misses being the full deal.

Minor Comments

The authors might like to point out that unlikely applicability of vicariance-only explanations for organismal distribution was spelled out by Darwin in his second chapter on paleontology in the *Origin* (i.e. chapter 10 in the 6th edition): there is more similarity among biota from within an interval of geological time than between adjacent intervals of time. Amongst trilobites, those who pushed vicariance as the explanation for the earliest Cambrian trilobite distributions never seem to have considered what their model implied the distributions of groups occurring somewhat later. Take, for example, late Cambrian "saukiid" trilobites which have several members of the same genus distributed widely around the equatorial continents during the Cambrian, as is the case with many other forms. All these must have been, according to the vicariant view, the results of rampant convergence occurring independently at the same time.

Lns 164 – oddly the authors exclude India from the list of vicariant distributions which is odd because this sector of Gondwana shows some of the clearest problems with vicariant models.

Reviewer #2 (Remarks to the Author):

The authors table a body of evidence that conflicts with a cryptic period of trilobite history. While I am favourably disposed towards the publication of this work, I identify a few places where I think they've either overstated evidence that fits their hypothesis or too casually dismissed evidence that doesn't fit it.

For example, "Arthropod traces in the Terreneuvian" is forced to attribute every single instance of

Rusophycus and Cruziana from the Terreneuvian to some kind of arthropod that is not a trilobite. As the authors note, at least some Cambrian examples are convincingly made by non-trilobites (e.g., the cited Pratt paper), as is also the case for rare post-Palaeozoic or freshwater examples (e.g., Zonneveld et al. 2002). But it seems to be a leap of faith to have so many post-Terreneuvian examples of these trace fossils associated with trilobites but absolutely none of the Terreneuvian ones. The authors rather too conveniently shrug this off with a "Fuxianhuia has bog-standard trilobite legs" (it doesn't) and over-confidently assert that other arthropods "undoubtedly" filled the trilobite niche. I appreciate that Cruziana stratigraphy allows for different behaviours and different tracemakers being in play before and after 521 Ma, and am aware of the tendency for the pre-trilobitic Rusophycus to be a bit wider than the younger ones (Jensen et al. 2013, and the Von Laere thesis), but then Rusophycus bonnarensis is the same before and after trilobite body fossils appear.

The character polarity for absence versus presence of dorsal facial sutures around line 295 is described as "uncertain". Eutrilobita (incl. Emuellidae) share a highly peculiar conformation of the ecdysial lines on the head shield. To moult your cheeks along a line that always runs around the palpebral lobe and to pop off the visual surface of the eye is, in the broader scheme of arthropod ecdysial lines, a very specific and peculiar morphology (even Peter Ax described it as a "remarkable method of moulting"). A marginal ecdysial line is present in relevant outgroups. I think this polarity is a no-brainer.

Figure 2 comes across as cherry picking, selecting a single sclerite (admittedly the most character-rich one) from two specimens/species, letting us see they look really similar, and then either insinuating that their classification in different families is suspect or else this overall similarity means that the earliest trilobites everywhere all looked the same. The phylogenies of Paterson et al. (2019), in which Fallotaspidae and Bigotiniidae are usually separated by a few nodes, are in line with the classification. I'm not saying those phylogenetic trees cannot be challenged but the statement in line 254 that they "are suggested to be closely related" is not a very powerful phylogenetic counter-argument.

I'm not sure if it's intentional, but the authors have taken historical biogeography right back to Lyell and Darwin. The reliance on pumice rafts to explain organismal distribution is the same as Lyell and Darwin's reliance on birds' feet and bills. Darwin examined sand grains on birds' feet and found some as big as seeds. Even the argument that "Well, pumice rafts might be rare but given the expanse of geological time they become common... even inevitable" was precisely what Lyell and Darwin argued for duck's feet, gusts of wind, etc. I think they've been over-enthusiastic in throwing out the vicariant baby with the bathwater. The statement about the Atlantic only being a barrier because of distance (lines 218-220) and not because of its age is presented as a fact, but examples can be cited that point the finger at vicariance (e.g., Jurado-Rivera et al. 2017, Scientific Reports, cave shrimps).

Reviewers' comments:

Reviewer #1 (Remarks to the Author):

Review of: Reassessing a cryptic history of early trilobite evolution
Authors: Holmes and Budd

Reviewer: Nigel Hughes

Overall recommendation: Accept with relatively modest modification

This is a well written piece of general interest to evolutionary biologists because it touches on an important general issue – vicariance vs dispersal, and is specifically aimed at an important issue in paleontology – the tempo and mode of the Cambrian radiation. I have long been sympathetic to the general argument made herein and I think the authors have done a good job of marshaling their various arguments in a cogent and interesting way that's an easy read for the non-specialist. I might imagine this piece fits well under the title of "perspective".

Major comments

That fallotaspidooids and bigotinids are close relatives is nicely illustrated in Fig 2, as is the discussion about the anterior facial suture in the Profallotaspis. I can accept this argument. However, the editor will have to judge whether this illustration alone is enough to make the point sufficiently for a "perspective" piece. I don't think it would be enough on its own for a research report, in which we'd want to see the arguments made in the text supported by a phylogenetic analysis. I think the paper would certainly be stronger if a character-based analysis could be presented or referred to. Without it the paper seems to go 90% of the way, but misses being the full deal.

We have now included a simplified version of the UCLN clock tree from Paterson et al. (2019; PNAS) and discuss this in the text. Although the topologies of the trees published in this paper vary somewhat, they are not inconsistent with our suggestion of a close relationship between fallotaspidooids and bigotinids (and a relatively accurate fossil record) and we have now discussed this in the text.

Minor Comments

The authors might like to point out that unlikely applicability of vicariance-only explanations for organismal distribution was spelled out by Darwin in his second chapter on paleontology in the Origin (i.e. chapter 10 in the 6th edition): there is more similarity among biota from within an interval of geological time than between adjacent intervals of time. Amongst trilobites, those who pushed vicariance as the explanation for the earliest Cambrian trilobite distributions never seem to considered what their model implied the distributions of groups occurring somewhat later. Take, for examples, late Cambrian "saukiid" trilobites which have several members of the same genus distributed widely around the equatorial continents during the Cambrian, as is the case with many other forms. All these must have been, according to the vicariant view, the results of rampant convergence occurring independently at the same time.

As both reviewers have pointed out the Darwin connection we have added additional text discussing this at the start of the vicariance section. In particular, we have included the above suggestion, and also discuss how Darwin pushed the idea of rare dispersal occurring regularly on geological timescales (mentioned by the other reviewer).

We have also discussed in more detail the patterns of diversification through time, in relation to the phylogeny mentioned above. These points and the phylogenetic discussion form the new section "Diversification through time and morpho(space)".

We also agree that there are many examples that could be used to highlight the problems with the trilobite vicariance model, although to keep the paper a reasonable length we have tried to focus on early Cambrian examples associated with the initial radiation. These patterns could certainly be discussed in much greater detail in another paper!

Lns 164 – oddly the authors exclude India from the list of vicariant distributions which is odd because this sector of Gondwana shows some of the clearest problems with vicariant models.

This was an error that I'm surprised neither of us picked up! We have now added.

Reviewer #2 (Remarks to the Author):

The authors table a body of evidence that conflicts with a cryptic period of trilobite history. While I am favourably disposed towards the publication of this work, I identify a few places where I think they've either overstated evidence that fits their hypothesis or too casually dismissed evidence that doesn't fit it.

For example, "Arthropod traces in the Terreneuvian" is forced to attribute every single instance of *Rusophycus* and *Cruziana* from the Terreneuvian to some kind of arthropod that is not a trilobite. As the authors note, at least some Cambrian examples are convincingly made by non-trilobites (e.g., the cited Pratt paper), as is also the case for rare post-Palaeozoic or freshwater examples (e.g., Zonneveld et al. 2002). But it seems to be a leap of faith to have so many post-Terreneuvian examples of these trace fossils associated with trilobites but absolutely none of the Terreneuvian ones. The authors rather too conveniently shrug this off with a "Fuxianhuia has bog-standard trilobite legs" (it doesn't) and over-confidently assert that other arthropods "undoubtedly" filled the trilobite niche. I appreciate that *Cruziana* stratigraphy allows for different behaviours and different tracemakers being in play before and after 521 Ma, and am aware of the tendency for the pre-trilobitic *Rusophycus* to be a bit wider than the younger ones (Jensen et al. 2013, and the Von Laere thesis), but then *Rusophycus bonnarensis* is the same before and after trilobite body fossils appear.

The main point that we are trying to make here is simply that the trace fossil record is not incongruent with a relatively accurate fossil record. Could pre-521 Ma trilobites have made these traces? Yes (although we think it unlikely given the other evidence). Could they be solely produced by closely related non-trilobite arthropods? Also yes (we consider this to be more likely).

The point you bring up about associations is an interesting one. Given that all trilobites should be primitively biomineralized, you could also interpret the lack of these associations in the Terreneuvian as evidence AGAINST their being made by trilobites. In other words, perhaps it shows that these traces are made by a more inclusive group of organisms that crosses this boundary.

The non-trilobite arthropods in particular have very similar overall/appendage morphologies to trilobites, and could fill this role. Obviously Arthropoda has a deeper history than the 521 Ma appearance of trilobites. Therefore, a possible explanation for 'trilobite-like' Terreneuvian traces is the large diversity of trilobite-like, non-mineralized euarthropods with trilobite-like appendages (other arthropods in particular) that must have been present across this period. Given they are present both before/after 521 Ma you would also expect these to be seen across this boundary, as with the *Rusophycus* example you point out. The mention of fuxianhuids and *Parapeytoia* is simply to suggest that even more basal taxa have at least reasonably similar appendages (i.e. biramous appendages with gnathobases).

Additionally, Donovan (2010; Lethaia) also made the point that we don't consider Triassic examples like the ones you mention above as evidence for post-Permian trilobites. Why should we consider their presence prior to 521 Ma in a different light when there are other candidates that could produce these?

We have modified this section substantially to make this more clear (including replacing the 'undoubtedly' you mentioned with 'likely'—we have also moderated similar points elsewhere). We have also added a new figure of a simplified arthropod phylogeny (based on fig. 6b of Ortega-Hernandez et al. 2013) to illustrate the pre-521 Ma history of this group more clearly (Fig. 2).

The character polarity for absence versus presence of dorsal facial sutures around line 295 is described as "uncertain". Eutriloibita (incl. Emuellidae) share a highly peculiar conformation of the ecdysial lines on the head shield. To moult your cheeks along a line that always runs around the palpebral lobe and to pop off the visual surface of the eye is, in the broader scheme of arthropod ecdysial lines, a very specific and peculiar morphology (even Peter Ax described it as a "remarkable method of moulting"). A marginal ecdysial line is present in relevant outgroups. I think this polarity is a no-brainer.

We were just being cautious here, but we now state this more clearly. We agree that this is the most likely alternative, and we already essentially state this in the next sentence by using *Profallotaspis* as an example to suggest how dorsal sutures might have evolved from a fallotaspidoid-type ancestor. Having said that, given that circumocular sutures are apparently shared by ALL trilobites, it does not seem impossible to independently develop (or lose) the anterior/posterior branches of the facial sutures given that the 'peculiar' aspect of such a system is already present.

Figure 2 comes across as cherry picking, selecting a single sclerite (admittedly the most character-rich one) from two specimens/species, letting us see they look really similar, and then either insinuating that their classification in different families is suspect or else this overall similarity means that the earliest trilobites everywhere all looked the same. The phylogenies of Paterson et al. (2019), in which Fallotaspidae and Bigotiniidae are usually separated by a few nodes, are in line with the classification. I'm not saying those phylogenetic trees cannot be

challenged but the statement in line 254 that they “are suggested to be closely related” is not a very powerful phylogenetic counter-argument.

We agree that this could be discussed more clearly in a phylogenetic context so we have included a simplified version of the Paterson et al. (2019) UCLN clock tree to illustrate our hypothesis. This now forms the major part of a new Figure 3, and some of the images from the previous Figure 2 have been relegated to a more minor part of this (we have also incorporated the taxonomy from the previous ‘Box 1’ into this figure). Note we have also modified Fig. 1 so it can be cited earlier and provide an overview of the faunal provinces.

The Paterson et al (2019) trees were rooted on the three fallotaspid taxa included in the analysis, although the oldest of these is dated at c. 519 Ma. Given our assumption of a relatively accurate fossil record (and that lack of dorsal sutures is probably primitive), we prefer to root the tree closer to a morphology reflecting the oldest fallotaspidoids (i.e. a Profallotaspis or Repinaella-type trilobite). Given these seem to be intermediate between fallotaspidids and bigotinids, we root the tree between these, and thus between the two major redlichiid suborders (the Olenellina and Redlichiina). The resulting topology illustrates how early olenellines and redlichiines could have rapidly evolved from a fallotaspidoid (potentially Profallotaspis-like) ancestral morphology (the original rooting is also compatible with this view).

I’m not sure if it’s intentional, but the authors have taken historical biogeography right back to Lyell and Darwin. The reliance on pumice rafts to explain organismal distribution is the same as Lyell and Darwin’s reliance on birds’ feet and bills. Darwin examined sand grains on birds’ feet and found some as big as seeds. Even the argument that “Well, pumice rafts might be rare but given the expanse of geological time they become common... even inevitable” was precisely what Lyell and Darwin argued for duck’s feet, gusts of wind, etc. I think they’ve been over-enthusiastic in throwing out the vicariant baby with the bathwater. The statement about the Atlantic only being a barrier because of distance (lines 218-220) and not because of its age is presented as a fact, but examples can be cited that point the finger at vicariance (e.g., Jurado-Rivera et al. 2017, Scientific Reports, cave shrimps).

As mentioned above, we have now discussed the Darwin connection in more detail. Obviously long-distance dispersal is important on geological timescales—island biogeographers have always known this (e.g. Cowie & Holland [2006] Dispersal is fundamental to biogeography and the evolution of biodiversity on oceanic islands, J. Biogeogr. 33, 193–198).

As you say, there is undoubtedly vicariant signal in some of the terrestrial and freshwater examples (particularly in terrestrial animals), and we now state and cite this more clearly. We are certainly not trying to say that vicariance is not an important factor. We are trying to make two main points here in the two sections:

1. That we can no longer simply assume that vicariance is the major factor (as was once generally thought, including when the trilobite hypotheses were first suggested), and models that do not take dispersal into account are no longer considered viable.
2. Regardless of the first point, marine invertebrate distributions do not show vicariant patterns associated with continental separation (we do see vicariance when populations become separated by land). The sentence you mention about the Atlantic is specifically referring to marine invertebrate taxa. Basically, it seems that marine taxa can disperse too easily for this to happen. In other words, invoking vicariance by continental separation to explain trilobite distributions is suggesting something that we do not observe in the modern world.

Based on these two points, it is highly unlikely that trilobite distributions were the result of vicariance by continental separation. We have edited these sections to make these points clearer.

Note that we state that: “*Disjunct distributions will always be controlled to some extent by tectonic movements, but this recent work suggests that any vicariant signal is often overlaid to a greater-or-lesser extent by instances of oceanic dispersal, creating a complex web of patterns that varies in different groups.*”). The cave shrimp example you cite shows exactly this, with both vicariance (potentially associated with the opening of the Atlantic) and dispersal suggested to play a role. However, these taxa are essentially restricted to aquatic habitats (although their larvae are salt tolerant, potentially explaining the instances of dispersal).

REVIEWERS' COMMENTS:

Reviewer #2 (Remarks to the Author):

Please remove the question mark from the tree in Figure 2 and edit the caption accordingly. Readers have had decades to learn how cladograms work and will understand that ((Conciliterga + Trilobita) Nektaspida) is simply the resolution of a three-item statement and does not state or even imply that there cannot be other unsampled lineages that are more closely related to trilobites than to concilitergans. One could interpolate any number of question marks on all branches of all trees but we don't do that.

I like the revised Figure 3, with the tree and photos combined, and am fine with how you have re-rooted. I see you've doubled down and reiterated Gerd Geyer's suggestion that fallotaspids and bigotinids are "likely closely related" on line 305 after already using it as the basis for your phylogenetics in line 271. Saying it twice doesn't add to its robustness; say it once. I accept that the intuition and gut feeling of a very experienced expert in early trilobites is worth a serious hearing but at the end of the day you're betting the house on an aside in the text of a paper rather than the tree you depict in Figure 3, which forces every species along the branches separating these two lineages to look exactly the same as in your photos.

Greg Edgecombe

Reviewers' comments:

Reviewer #2 (Remarks to the Author):

Please remove the question mark from the tree in Figure 2 and edit the caption accordingly. Readers have had decades to learn how cladograms work and will understand that ((Conciliterga + Trilobita) Nektaspida) is simply the resolution of a three-item statement and does not state or even imply that there cannot be other unsampled lineages that are more closely related to trilobites than to concilitergans. One could interpolate any number of question marks on all branches of all trees but we don't do that.

We have removed the question mark and the reference to this in the figure caption.

I like the revised Figure 3, with the tree and photos combined, and am fine with how you have re-rooted. I see you've doubled down and reiterated Gerd Geyer's suggestion that fallotaspidids and bigotinids are "likely closely related" on line 305 after already using it as the basis for your phylogenetics in line 271. Saying it twice doesn't add to its robustness; say it once. I accept that the intuition and gut feeling of a very experienced expert in early trilobites is worth a serious hearing but at the end of the day you're betting the house on an aside in the text of a paper rather than the tree you depict in Figure 3, which forces every species along the branches separating these two lineages to look exactly the same as in your photos.

We have removed the second reference at line 305 as requested, although it should be pointed out that we are not basing this argument purely on what Gerd has said. We are arguing for a close relationship based on obvious morphological similarities between these groups and pointing out that such a view is not incompatible with the phylogeny. We are mainly referencing Gerd to point out that this is not a new idea and to give him due credit.